# Inflammatory profiles are associated with long COVID up to 6 months after COVID-19 onset: A prospective cohort study of individuals with mild to critical COVID-19

Elke Wynberg[1,2]☉*, Alvin X. Han[1]☉, Hugo D. G. van Willigen[1]☉, Anouk Verveen[3]☉, Lisa van Pul[4]☉, Irma Maurer[4]☉, Ester M. van Leeuwen[4]☉, Joost G. van den Aardweg[5], Menno D. de Jong[1], Pythia Nieuwkerk[3], Maria Prins[2,6], Neeltje A. Kootstra[3]☉, Godelieve J. de Bree[6]☉, on behalf of the RECoVERED Study Group¶

1 Department of Medical Microbiology & Infection Prevention, Amsterdam UMC, University of Amsterdam, Amsterdam Institute for Infection and Immunity, Amsterdam, the Netherlands, 2 Department of Infectious Diseases, Public Health Service of Amsterdam, Amsterdam, the Netherlands, 3 Department of Medical Psychology, Amsterdam UMC, Amsterdam Public Health Research Institute, University of Amsterdam, Amsterdam, the Netherlands, 4 Department of Experimental Immunology, Amsterdam UMC, University of Amsterdam, Amsterdam Institute for Infection and Immunity, Amsterdam, the Netherlands, 5 Department of Pulmonology, Amsterdam UMC, University of Amsterdam, Amsterdam, the Netherlands, 6 Department of Infectious Diseases, Internal Medicine, Amsterdam UMC, University of Amsterdam, Amsterdam Institute for Infection and Immunity, Amsterdam, the Netherlands

☉ These authors contributed equally to this work.
¶ Membership of the RECoVERED study group is listed in the Acknowledgments.
* elke.wynberg@gmail.com

**Data Availability Statement:** Data cannot be shared publicly because of confidentiality

## Abstract

### Background

After initial COVID-19, immune dysregulation may persist and drive post-acute sequelae of COVID-19 (PASC). We described longitudinal trajectories of cytokines in adults up to 6 months following SARS-CoV-2 infection and explored early predictors of PASC.

### Methods

RECoVERED is a prospective cohort of individuals with laboratory-confirmed SARS-CoV-2 infection between May 2020 and June 2021 in Amsterdam, the Netherlands. Serum was collected at weeks 4, 12 and 24 of follow-up. Monthly symptom questionnaires were completed from month 2 after COVID-19 onset onwards; lung diffusion capacity ($D_{LCO}$) was tested at 6 months. Cytokine concentrations were analysed by human magnetic Luminex screening assay. We used a linear mixed-effects model to study log-concentrations of cytokines over time, assessing their association with socio-demographic and clinical characteristics that were included in the model as fixed effects.

### Results

186/349 (53%) participants had ≥2 serum samples and were included in current analyses. Of these, 101/186 (54%: 45/101[45%] female, median age 55 years [IQR = 45–64]) reported

agreements. Relevant data can be made available upon request via the RECoVERED Study Steering Committee. Please contact the AMC Medical Ethical Committee for additional information: https://metc.amsterdamumc.org/contact/.

**Funding:** This work was supported by the Netherlands Organization for Health Research and Development (ZonMw) [101500062010002 to M.D. d.J. and 10430072110003 to G.J.d.B] and the Public Health Service of Amsterdam [R&D grants in 2021 and 2022 to M.P.]. The funders had no role in study design, data collection, data analysis, data interpretation or data reporting. ZonMw website: https://projecten.zonmw.nl/nl/project/recovered GGD Amsterdam website: https://www.ggd. amsterdam.nl/ggd/.

**Competing interests:** The authors have declared that no competing interests exist.

PASC at 12 and 24 weeks after COVID-19 onset. We included 37 reference samples (17/37 [46%] female, median age 49 years [IQR = 40–56]). In a multivariate model, PASC was associated with raised CRP and abnormal diffusion capacity with raised IL10, IL17, IL6, IP10 and TNFα at 24 weeks. Early (0–4 week) IL-1β and BMI at COVID-19 onset were predictive of PASC at 24 weeks.

## Conclusions

Our findings indicate that immune dysregulation plays an important role in PASC pathogenesis, especially among individuals with reduced pulmonary function. Early IL-1β shows promise as a predictor of PASC.

## Background

Almost half of individuals [1] infected with severe acute respiratory syndrome coronavirus 2 (SARS-CoV-2) are estimated to experience symptoms related to "long COVID" or post-acute sequelae of COVID-19 (PASC). PASC has been defined by the World Health Organization [2] as symptoms lasting longer than 3 months after initial infection, with these symptoms lasting for at least 2 months with no other explanation. The symptoms associated with PASC involve numerous different organ systems, including cough (respiratory), palpitations (cardiovascular), post-exertional malaise (musculoskeletal), and sleep disturbances, reduced concentration and cognitive fatigue (neuropsychiatric) [3]. Similar sequelae have been reported in children, although studies in paediatric populations are scarce [4].

There are many uncertainties regarding the pathophysiology of PASC, with hypotheses including viral persistence, hypercoagulability, autonomic dysfunction, chronic inflammation [3]–or a combination of these. Several studies have suggested that hyperinflammation observed during acute infection among individuals with severe COVID-19 [5–7] may persist among those with ongoing symptoms [8]. However, whether chronic inflammation also underpins the pathogenesis of PASC in individuals with initially mild or moderate COVID-19 is less clear. Studies investigating chronic inflammation in PASC to date have been largely inconclusive due to heterogeneity in research objectives, study design and population. Some studies aimed to outline immune disturbances in the first months after infection [9–11] in order to better understand COVID-19 immunopathology, whilst others specifically aimed to identify early inflammatory predictors for PASC that could be of clinical relevance [12–16]. As such, prospective data that aims to both outline longitudinal immune profiles among individuals with mild to critical COVID-19, and identify possible early markers of PASC, is needed to better understand the role of chronic inflammation in the development and persistence of PASC.

Using data from the prospective RECoVERED cohort study in Amsterdam, the Netherlands, we firstly aimed to investigate the evolution of serum levels of specific cytokines from COVID-19 onset onwards among individuals with PASC, compared to individuals without PASC. Reference samples were included from SARS-CoV-2-naïve individuals. Secondly, we aimed to explore the determinants of cytokine levels at 3 and 6 months following COVID-19 onset, including PASC status. Finally, aimed to identify possible early biomarkers for the development of PASC.

## Methods

We present the methodology of our cohort study according to The Strengthening the Reporting of Observational Studies in Epidemiology (STROBE) guidelines for reporting observational studies [17].

### Study design and participant enrolment

RECoVERED is a prospective cohort study of adults aged 16–85 with SARS-CoV-2 infection between May 2020 and June 2021 in Amsterdam, the Netherlands. Full details of study procedures and additional inclusion/exclusion criteria have been published elsewhere [18]. Briefly, participants were enrolled within 7 days of diagnosis (at the Public Health Service of Amsterdam) or hospital admission (at the Amsterdam University Medical Centres [UMC]). All participants had laboratory-confirmed SARS-CoV-2 infection. RECoVERED was approved by the medical ethical review board of the Amsterdam University Medical Centres (METC NL73759.018.20). All participants provided written informed consent.

For the current analysis, we included a sub-set of participants with samples from at least two different time-points and with follow-up of at least 3 months following SARS-CoV-2 infection up to November 2022. We additionally included reference samples from SARS-CoV-2-uninfected, healthy individuals (i.e., no comorbidities) collected between March 2020 and November 2021, in order to better understand the impact of SARS-CoV-2 infection on absolute levels of cytokines. These samples were randomly selected from a prospective serologic surveillance cohort study among hospital healthcare workers in the Amsterdam UMC (S3 study; METC NL73478.029.20) [19].

### Data collection

During the first month of follow-up, trained study staff interviewed participants on the presence of 21 different COVID-19 symptoms, took physical measurements, and recorded participants' past medical and socio-demographic characteristics. Recorded symptoms included: fatigue, cough, fever, rhinorrhoea, sore throat, dyspnoea, loss of smell and/or taste, chest pain, headache, abdominal pain, confusion, arthralgia, myalgia, loss of appetite, wheeze, skin rash, nausea and/or vomiting, diarrhoea, earache, and spontaneous bleeding. Between months 3 to 12 of follow-up, monthly online questionnaires on the presence of the same 21 COVID-19 symptoms were completed by participants. Lung function tests including diffusion capacity ($D_{LCO}$) were performed at 1, 6 and 12 months after COVID-19 onset (detailed methods described elsewhere [20]).

Serum samples were collected at day 0 and 7 and subsequently at months 1, 3, and 6 of follow-up. Additional serum samples were collected per protocol within 24 hours of receiving a COVID-19 vaccination and 7 and 28 days following each COVID-19 vaccination. All serum samples were processed within 24 hours and stored at -80˚C. In the present study, we defined three time-frames of sample collection for longitudinal analyses: 0–4, 9–12 and 21–24 weeks after COVID-19 onset. Post-vaccination samples were defined as those collected within 28 days after administration of a COVID-19 vaccine.

C-reactive protein (CRP), monocyte/macrophage surface receptors (CD14, CD163), tumor necrosis factor (TNF)-α, interferon-γ-inducible protein 10 (IP-10)/CXCL10, monocyte chemoattractant protein (MCP)1/CCL2, and interleukins (IL)1β, IL2, IL6, IL10, IL13, and IL17A concentrations were analyzed in serum by human magnetic luminex screening assay (LXSAHM-02 and LXSAHM-10; R&D Systems) using the Bio-Plex 200 System (Bio-Rad Laboratories). Assays were performed according to the manufacturer's instructions. Titers of total antinuclear antibodies (ANA) as markers of auto-immunity were determined by

immunofluorescence using the HEp-20-10 test kit (Euroimmun AG) for samples collected ≤4 weeks after COVID-19 onset.

## Outcomes

The primary outcomes of our analysis were cytokine levels at 9–12 and 21–24 weeks after COVID-19 onset. Our secondary outcomes were: PASC status at 6 months, and CRP and IL6 levels at 21–24 weeks.

## Definitions

COVID-19 onset was defined as the earliest day that COVID-19 symptoms were experienced for symptomatic patients, or the date of SARS-CoV-2 diagnosis for asymptomatic patients. PASC was defined as reporting at least one COVID-19 symptom that, from COVID-19 onset, occurred within one month and continued beyond 12 weeks [21]; symptoms arising after one month from COVID-19 onset were not attributed to PASC. COVID-19 clinical severity was categorised according to WHO criteria [22]: mild disease was defined as having a RR <20/ minute and SpO2>94% on room air at both D0 and D7 study visits; moderate disease as having a RR20–30/minute and SpO2 90–94% on room air (or receiving oxygen therapy, if no off-oxygen measurement available) at either visit; severe disease as having a RR >30/minute and SpO2<90% on room air (or receiving oxygen therapy) at either visit; critical disease as requiring ICU admission as a result of COVID-19 at any point. BMI was coded in $kg/m^2$ as: <25, underweight or normal weight; 25–29, overweight; ≥30, obese. Diffusion capacity ($D_{LCO}$) at 6 months after COVID-19 onset was defined as impaired according to American Thoracic Society/European Respiratory Society guidelines [23], as described previously [20].

## Statistical analysis

In the first descriptive analysis, socio-demographic, clinical and study characteristics were compared between included participants with and without PASC at 12 weeks after COVID-19 onset. Continuous variables were analyzed using the Kruskal-Wallis test and categorical and binary variables were compared using the Pearson $\chi^2$ test (or Fisher exact test if n <5). To assess selection bias, we compared features of included and excluded participants. We used the Fisher's exact test to evaluate the association between PASC and impaired diffusion capacity at 6 months after COVID-19 onset.

The primary outcome of the study was cytokine levels at 9–12 and 21–24 weeks after COVID-19 onset. A correlation matrix (Pearson's correlation coefficient) of cytokines at 0–4, 9–12 and 21–24 weeks were used to help interpret the complexity of inter-marker associations. Next, we determined box and whisker plots of of median (IQR) log-concentrations of cytokines at 9–12 and 21–24 weeks after COVID-19 onset. During this analysis, data of study participants with and without PASC at 12 and 24 weeks after COVID-onset were compared with data from the uninfected healthy control population using the Mann-Whitney and Bonferroni correction tests. Comparisons were first performed using data from the whole cohort (mild to critical COVID-19) and subsequently restricted to participants with mild or moderate COVID-19. To additionally assess differences in cytokine levels using an objective measure of PASC, we compared median (IQR) log-concentrations of cytokines measured at 21–24 weeks between participants with and without an impaired diffusion capacity ($D_{LCO}$) at 6 months after COVID-19 onset. We subsequently used linear mixed-effects tests to assess the effect of pre-COVID and COVID-related factors (including PASC status) on cytokine concentrations in two cross-sectional analyses. To this end we applied two linear multivariate mixed-effects models: the first at 3 months (serum collected at 9–12 weeks) and the second at 6 months

(serum collected at 21–24 weeks) after COVID-19 onset. We modelled log concentrations of cytokines with time since COVID-19 onset as a random effect (to account for variability in the timing of serum sampling around the 3- and 6-month time points) and as fixed effects: sex, age and clinical characteristics (i.e. body mass index [BMI] and comorbidities, defined at COVID-19 onset), PASC status (at 12 and 24 weeks after COVID-19 onset in each model, respectively), and recent (<4 weeks) vaccination. The selection of effects was chosen following published risk factors for long COVID (reviewed by [3]). We substituted PASC status (based on self-reported symptoms) for impaired diffusion capacity ($D_{LCO}$) in an additional model at 6 months after COVID-19 onset. Condition indices were computed to ensure that there was no collinearity among the variables (i.e., condition index<10).

The secondary outcome was the identification of predictive determinants at COVID-19 onset of later PASC. We performed a random forest regression, a model-free machine-learning approach, to identify the early predictors of: (1) PASC at 24 weeks and (2) higher levels, at 21–24 weeks after COVID-19 onset, of CRP and IL-6. We performed k-fold cross validation (k = 5) to tune the hyperparameters of each random forest regressor. We used F1 scores and mean squared error as scoring functions of the random forest regressors used in (1) and (2)/(3) respectively. We then computed Shapley additive explanation values as measures of importance for the different predictors [24]. CRP and IL6 at 0–4 weeks were not individually included as predictors of their measurements at 21–24 weeks.

All data were collected and stored in a secure database and exported for statistical analyses to Python. Analyses were performed using the statsmodels package (v. 0.13.2) [25], whilst the random forest regression analyses were performed in Python using the scikit-learn package (v. 1.1.3).

## Results

### Baseline characteristics of the study population

Of 349 RECoVERED participants, 186 (53.3%) had at least two serum sampling moments and were included in the present study. Included participants were more likely to be male and have initially mild COVID-19 (S1 Table in S1 File). 101/186 (54%; 45/101[45%] female, median age 55 years [IQR = 45–64]) included participants reported PASC at 12 weeks after COVID-19 onset (Table 1 and S2 Table in S1 File), of whom none reported their symptoms resolving or were lost to follow-up between weeks 12 and 24. A subgroup (72/186; 38.7%) of participants had a diffusion capacity at 6 months after COVID-19 onset. Among the 22/72 who exhibited impaired diffusion capacity, more than half (12/22; 54.5%) also reported PASC at 6 months (p = 0.031).

We included reference samples from 37 individuals with no history of SARS-CoV-2 infection (median (IQR) age 49 years [IQR = 39.5–55.5]; 17/37 [45,9%] female). Seven (18.9%) reference samples were collected within 6 months after a primary COVID-19 vaccination series.

### Primary outcome: Cytokine levels at 9–12 and 21–24 weeks after COVID-19 onset and their determinants

S1 Fig and S3 Table in S1 File demonstrate aberrant cytokine levels in RECoVERED study participants, both with and without PASC, compared to the reference group. Cytokine correlation matrices are shown in S2 Fig in S1 File. In univariable analyses, individuals with PASC tended to have lower levels of sCD14, IL10, IL17, IL1β, IL6 and TNFα (Fig 1) compared to participants without PASC at 9–12 weeks after COVID-19 onset. By 21–24 weeks, participants with PASC had significantly higher concentrations of IL10, IL1β and sCD14 than those without

**Table 1. Characteristics of study participants with and without PASC at 12 weeks after COVID-19 onset.**

| | Total | Symptoms resolved within 12 weeks (no PASC) | Ongoing symptoms at 12 weeks (PASC) | p-value* |
|---|---|---|---|---|
| | N = 186 | N = 85 | N = 101 | |
| Sex | | | | 0.022 |
| Male | 117 (63%) | 61 (72%) | 56 (55%) | |
| Female | 69 (37%) | 24 (28%) | 45 (45%) | |
| Age, years | 52.0 (37.0–62.0) | 48.0 (33.0–60.0) | 55.0 (45.0–64.0) | 0.006 |
| BMI, kg/m$^2$ | 25.7 (23.2–29.3) | 25.4 (23.1–27.6) | 26.2 (23.2–29.7) | 0.16 |
| ANA positive ** | | 8 (14%) | 7 (18%) | - |
| Clinical severity score | | | | <0.001 |
| Mild | 64 (34%) | 44 (52%) | 20 (20%) | |
| Moderate | 79 (42%) | 33 (39%) | 46 (46%) | |
| Severe/critical | 43 (23%) | 8 (9%) | 35 (35%) | |
| COVID-19 vaccination status (primary series) | | | | 0.27 |
| Not vaccinated | 1 (1%) | 1 (1%) | 0 (0%) | |
| Vaccinated | 185 (99%) | 84 (99%) | 101 (100%) | |
| Time from COVID-19 onset to first vaccination, days | 244 (151–363) | 226 (142–303) | 270 (158–386) | 0.014 |

Abbreviations: BMI, body mass index; COVID-19, coronavirus disease 2019; HR, heart rate; ICU, intensive care unit; LTFU, lost to follow-up; OECD, Organisation for Economic Co-operation and Development; NA, not applicable; PCR, polymerase chain reaction; SpO2, oxygen saturation on room air; RR, respiratory rate; SARS-CoV-2, severe acute respiratory syndrome coronavirus 2.

* Continuous variables presented as median (IQR) and compared using the Kruskal-Wallis test; categorical and binary variables presented as n(%) and compared using the Pearson $\chi^2$ test (or Fisher exact test if n <5).

Clinical severity groups defined as: mild as having an RR <20/min and SpO2 on room air >94% at both D0 and D7; moderate disease as having a RR 20–30/minutes, SpO2 90–94% and/or receiving oxygen therapy at D0 or D7; severe disease as having a RR >30/minutes or SpO2 <90% at D0 or D7; critical disease as requiring ICU admission. COVID-related comorbidities are based on WHO Clinical Management Guidelines and include: cardiovascular disease (including hypertension), chronic pulmonary disease (excluding asthma), renal disease, liver disease, cancer, immunosuppression (excluding HIV, including previous organ transplantation), previous psychiatric COVID-19 and dementia. Physical measurements at D0 and D7 study visits. Oxygen saturation measured on room air if possible or retrieved from ambulance records for hospitalized participants admitted on oxygen on day of enrollment. Physical measurements not displayed for individuals with critical disease due to unreliability of measurements at admission for critically-ill patients.

** Of all participants with PASC there were 59 samples available for ANA testing, of all participants without PASC there were 39 samples available for ANA testing.

PASC. When restricting our analyses to participants with initially mild or moderate COVID-19, no difference in IL10 levels between participants with and without PASC remained at 21–24 weeks; however, the higher levels of IL1β and sCD14 observed among participants with PASC compared to those without PASC remained (S3 Fig in S1 File). Individuals with an impaired diffusion capacity (D$_{LCO}$) had significantly higher log-concentrations of IP10, IL10, IL6 and TNFα than participants with normal diffusion capacity at 21–24 weeks (S4 Fig in S1 File).

In multivariable analyses, participants with PASC had significantly lower levels of IL10 and TNF-α compared to participants who had recovered from their symptoms at 9–12 weeks. At 9–12 weeks after disease onset, participants who had initially severe COVID-19 tended to have significantly higher levels of IP10 and sCD163 and lower levels of IL10, IL6, TNFα, IL17 and IL13 (Fig 2) compared to those with mild or moderate disease, when adjusting for other covariates. Independent of initial disease severity, having received dexamethasone during acute COVID-19 was associated with higher levels of IL6, IL10, sCD14 and CRP compared to those

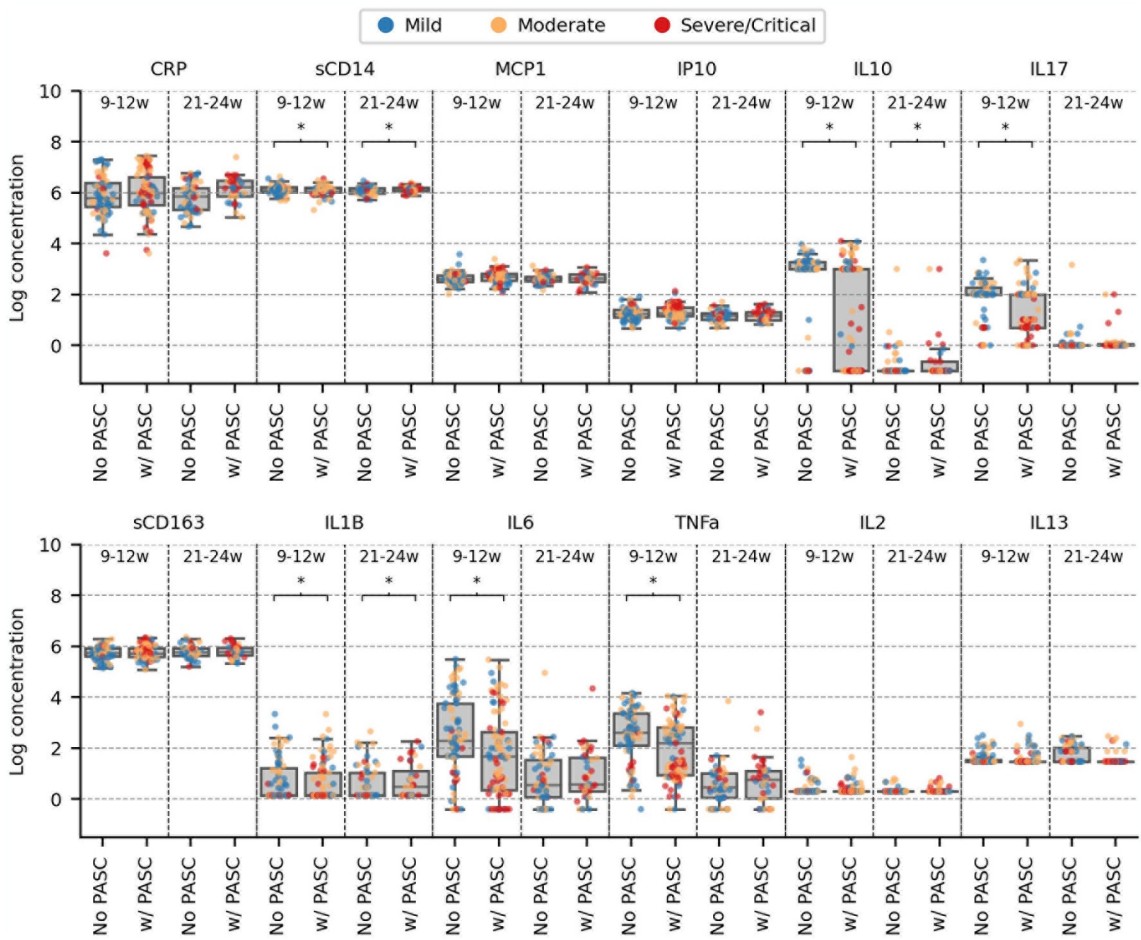

**Fig 1. Box and Whisker plots of serum cytokine levels in the study population at 9–12 and 21–24 weeks post-COVID.** Individuals were stratified by whether or not they reported post-acute sequelae of COVID-19 (PASC) at 12 (for measurements at 9–12 weeks) or 24 (for measurements at 21–24 weeks) after COVID-19 onset. Each dot represents an individual coloured by initial severity of COVID-19. The mean value is plotted for each individual with multiple measurements within the binned time period. A Mann-Whitney U test was used to test if measurements between those with and without PASC were significantly different. Multiple testing correction was performed using the Bonferroni method and comparisons with family-wise error rate < 0.05 were marked with an "*".

who did not receive dexamethasone, in multivariable analyses. Age $\geq$ 60 years and BMI $\geq$ 30 kg/m$^2$ at COVID-19 onset were associated with higher levels of IL2 and IP10, and MCP1 and CRP, respectively, when adjusting for other covariates.

By 21–24 weeks after COVID-19 onset, participants with ongoing PASC at 24 weeks after COVID-19 onset had higher concentrations of CRP in multivariable analyses compared to participants without PASC (Fig 3). Of note, when instead of PASC, impaired diffusion capacity was added to the analysis, we found that impaired diffusion capacity was associated with higher levels of CRP, IL6, TNFα, IP10, IL10 and IL17 in multivariable analyses (S5 Fig in S1 File). In addition, the effect of obesity and of older age on cytokine concentrations became more pronounced at 21–24 weeks compared to at 9–12 weeks (Fig 3). Individuals who received dexamethasone had significantly lower levels of TNFα, IL6 and IL1β by 21–24 weeks in multivariable analyses, in contrast to findings at 9–12 weeks after COVID-19 onset.

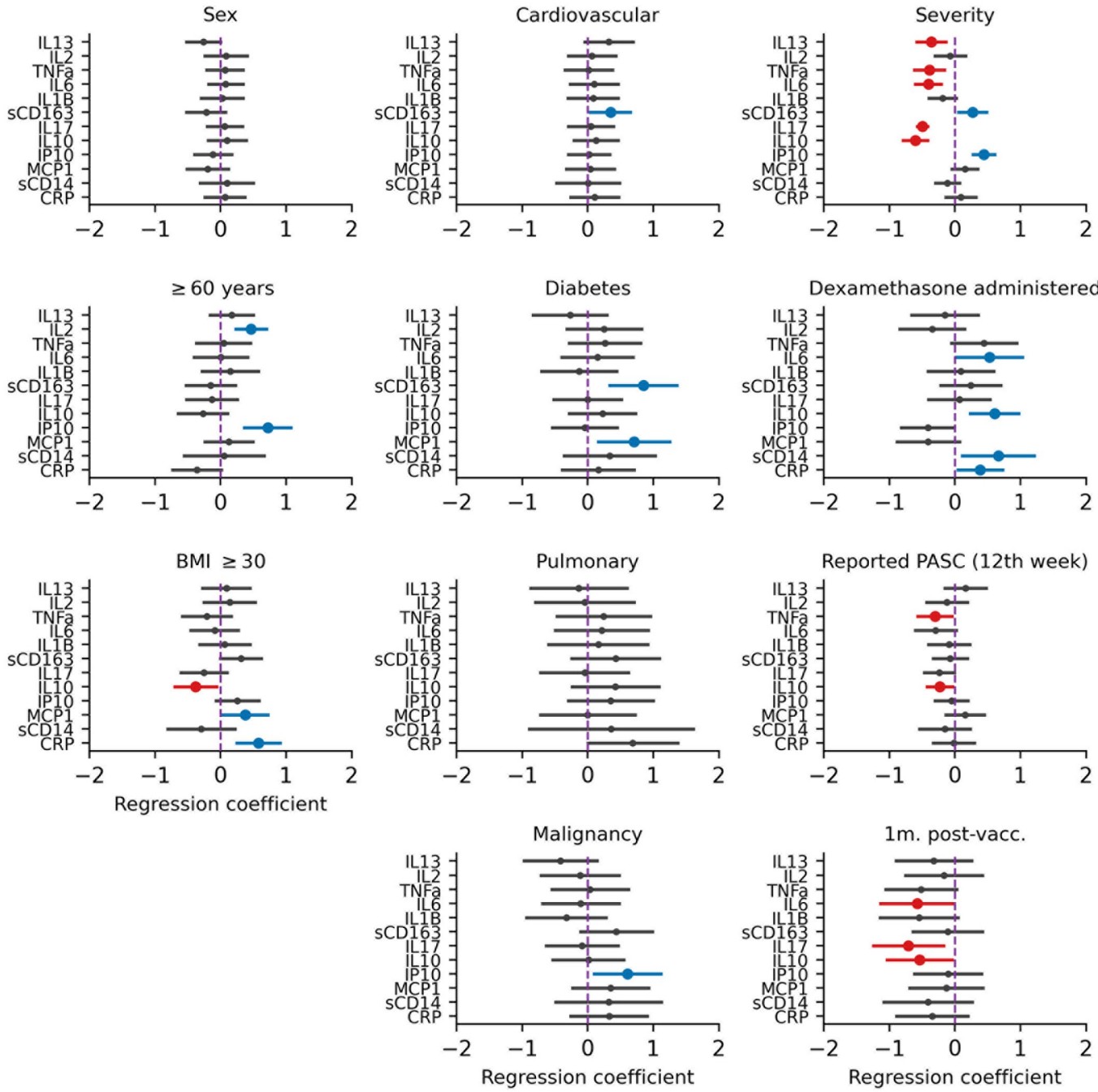

**Fig 2. Multivariate linear regression analysis of factors associated with inflammatory marker concentrations at 9–12 weeks after COVID-19 onset.** For each inflammatory marker, we performed a mixed effects linear regression of various factors: characteristics present prior to COVID-19 onset, COVID-19-related factors, and post-COVID-19 related factors. Characteristics present prior to COVID-19 onset included sex, age ($\geq$60 or <60 years old), body mass index (BMI) $\geq$ 30, and the presence of comorbidities (including cardiovascular disease, diabetes mellitus, chronic pulmonary disease or current cancer). COVID-19-related factors included the severity of initial COVID-19 disease, if dexamethasone was administered, and current (at 12 weeks) presence of post-acute sequelae of COVID-19 (PASC). Post-COVID-19 related factors included measurement of inflammatory markers within four weeks after SARS-CoV-2 vaccination. Statistically significant negative effects (associations with lower cytokine concentrations) are shown in red whilst positive effects (associations with higher cytokine levels) are shown in blue.

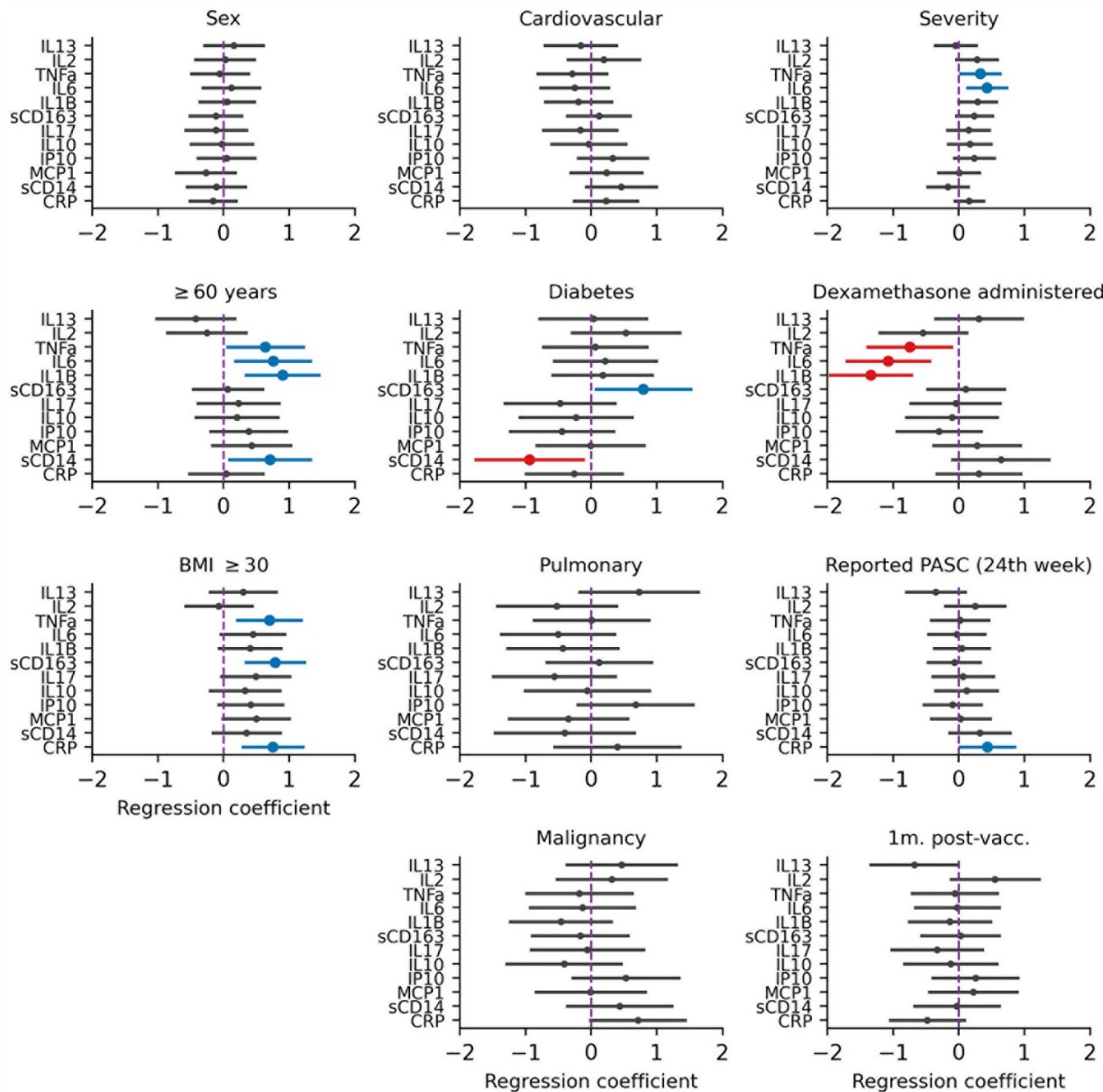

**Fig 3. Multivariate linear regression analysis of factors associated with inflammatory marker concentrations at 21–24 weeks after COVID-19 onset.** For each inflammatory marker, we performed a mixed effects linear regression of various factors: characteristics present prior to COVID-19 onset, COVID-19-related factors, and post-COVID-19 related factors. Characteristics present prior to COVID-19 onset included sex, age ($\geq 60$ or $<60$ years old), body mass index (BMI) $\geq 30$, and the presence of comorbidities (including cardiovascular disease, diabetes mellitus, chronic pulmonary disease or current cancer). COVID-19-related factors included the severity of initial COVID-19 disease, if dexamethasone was administered, and current (at 24 weeks) presence of post-acute sequelae of COVID-19 (PASC). Post-COVID-19 related factors included measurement of inflammatory markers within four weeks after SARS-CoV-2 vaccination. Statistically significant negative effects (associations with lower cytokine concentrations) are shown in red whilst positive effects (associations with higher cytokine levels) are shown in blue.

## Secondary outcome: Early predictors of PASC and ongoing inflammation 6 months after COVID-19 onset

We found early IL1β and BMI at COVID-19 onset to be the strongest predictors of PASC at 21–24 weeks (Fig 4a), using Shapley additive explanation values as measures of importance

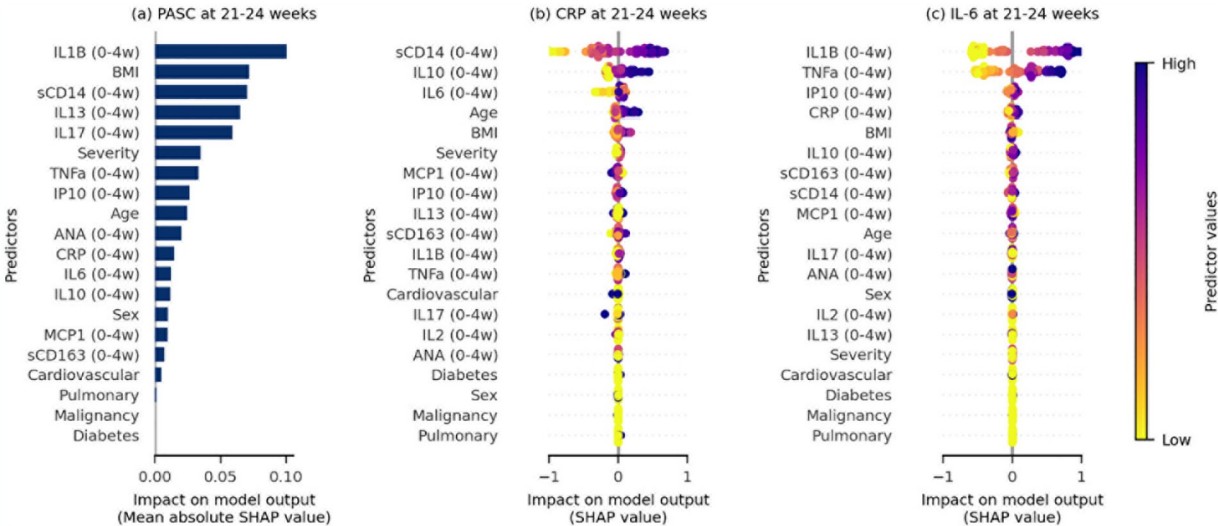

**Fig 4. Early (0–4 week) predictors of PASC and CRP/IL-6 levels at 21–24 weeks after COVID-19 onset.** Importance of different predictors (ordered from most [top] to least [bottom] important according to Shapley additive explanation (SHAP) values) on (a) individuals reporting PASC, level of (b) CRP and (c) IL-6 measurements at 21–24 weeks. Predictors include socio-demographic factors (i.e. age at infection, years, and sex), body mass index (BMI, kg/m$^2$), presence/absence of comorbidities (i.e. cardiovascular disease, diabetes mellitus, chronic pulmonary disease and current cancer [malignancy]) and mean log-concentrations of inflammatory markers measured at 0–4 weeks after COVID-19 onset. For (a), each horizontal bar denotes the mean absolute SHAP value associated with the predictor. The larger the mean absolute SHAP value, the more important the covariate is in predicting the outcome of reporting PASC at 21–24 weeks. For (b) and (c), each point is the SHAP value for the corresponding predictor for each individual. The color of each point represents the value of the predictor for an individual. For instance, in (b), sCD14 levels at 0–4 weeks is the most important predictor for levels of CRP at 21–24 weeks, with high mean levels of sCD14 at 0–4 weeks expected to yield a higher level of CRP at 21–24 weeks.

[24]. Higher levels of sCD14, and to a lesser extent IL10, at 0–4 weeks were important predictors of higher levels of CRP at 21–24 weeks (Fig 4b). IL1β and TNFα measurements at 0–4 weeks were key predictors of IL6 levels at 21–24 weeks (Fig 4c). Total ANA titers were less important than BMI, initial disease severity and age in the prediction of ongoing PASC at 21–24 weeks after COVID-19 onset, but more important than sex and early levels of CRP, IL6 and IL10 (Fig 4a).

## Discussion

In our prospective cohort study of participants with mild to critical COVID-19, we explored the evolution of cytokine concentrations from disease onset and the effect of PASC on cytokine concentrations up to 6 months after COVID-19 onset. Participants with PASC displayed significantly higher concentrations of pro-inflammatory CRP at 21–24 weeks after COVID-19 onset compared to those without PASC. In addition, when defining PASC as having an impaired diffusion capacity at 6 months after COVID-19 onset, we identified an association with raised levels of several other pro-inflammatory cytokines, also when adjusting for comorbidities and initial COVID-19 severity. This suggests that individuals with objectifiable pathology in the context of PASC may experience more pronounced immune dysregulation. Raised IL1β at 0–4 weeks after COVID-19 onset was strongly predictive of persistent PASC at 24 weeks after COVID-19 onset, and warrants further exploration as a possible predictive biomarker for PASC.

In our study, participants with PASC, as defined by self-reported symptoms, had raised CRP at 6 months after COVID-19 onset when adjusting for age, sex, comorbidities (including

obesity) at COVID-19 onset, initial COVID-19 severity, dexamethasone treatment and recent COVID-19 vaccination. We did not, however, identify increased levels of numerous other pro-inflammatory cytokines including IL6, IL1β and TNFα among individuals with PASC, as has been described in other studies [9, 16, 26–28]. Interestingly, however, when substituting self-reported PASC status with impaired diffusion capacity as a more objective measure, the more complex pattern of raised cytokines (including IL10, IL6, IL17, IP10 and TNFα) observed in other studies was echoed in our cohort, also when adjusting for possible confounders such as age and comorbidities. This observation may reflect two possible explanations. Firstly, that confirming PASC status with an objective measure increased the precision of the definition, reducing misclassification bias resulting from self-reported symptoms not related to PASC and thus allowing an association with pro-inflammatory cytokines to be revealed. Alternatively, PASC is an umbrella term for multiple conditions, within which individuals with measurable lung abnormalities experience a persistent hyperinflammatory process whilst the underlying cause of those with other symptom clusters [3, 29] cannot be explained by aberrant cytokines. Our findings thus suggest that immune dysregulation plays an important role in PASC pathogenesis in some individuals, and that those with measurable persistent pathology following COVID-19 may exhibit more pronounced hyperinflammation.

We also observed that higher levels of IL1β (and to a lesser extent sCD14, IL13, IL17 and TNFα) at 0–4 weeks were strongly associated with ongoing PASC at 24 weeks after COVID-19 onset. This is of interest because we did not find an association of self-reported PASC with IL1β at 21–24 weeks, suggesting that early IL1β induced tissue damage [30] or endothelial dysfunction [31] may predispose individuals to later symptomatology. Given that IL1β has been implicated as a marker of neuroinflammation [32] and profound neurological symptoms are part of PASC [33], further exploration of the role of this cytokine in PASC pathogenesis is warranted. In addition, we found that elevated early sCD14 and IL10 levels were most strongly predictive of raised CRP at 21–24 weeks, whilst IL1β and TNFα levels were strongly congruent with persistently elevated IL6 at 24 weeks. These findings help provide insight into the possible immunopathogenesis of PASC. First, the association between high sCD14 levels at 0–4 weeks and persistently elevated CRP and PASC at 24 weeks suggests the key role of monocyte-macrophage activation in the acute phase [34], driving long-term inflammation. Second, our findings imply that IL1β-mediated acute inflammation contributes to ongoing symptomatology and hyperinflammation many months after infection. Taken together, our data add to an increasing number of studies [26, 27] that observe inflammation as reflection of immune dysregulation in PASC. However, it currently remains unclear which factors (for instance ongoing antigen persistence [35]) may drive immune dysregulation, and if immune dysregulation is a result rather than a cause of PASC pathology [27]. Consistent with previous findings(34,35), higher BMI at COVID-19 onset was also an independent predictor of ongoing PASC at 24 weeks, also when adjusting for concentrations of numerous pro-inflammatory cytokines. This suggests that the effect of BMI on PASC risk is not only mediated by inflammation (as indicated by the association of obesity at COVID-19 onset with raised TNFα, sCD163 and CRP at 21–24 weeks), but also by additional physiological and metabolic processes [36]. It is crucial that ongoing studies on the role of inflammation in PASC pathogenesis account for the central role played by BMI.

Our study benefits from its prospective design, thus limiting selection bias, detailed symptom data, representation of a wide range of COVID-19 severity, and the uniformity of sample processing and analysis procedures conducted in a central laboratory. However, our study also has limitations. Firstly, we did not have symptom data pre-dating SARS-CoV-2 infection. We therefore cannot be certain that reported symptoms were a result of COVID-19 or due to pre-existing comorbidities, which may have resulted in misclassification bias. However, we

attempted to overcome this bias by defining long COVID symptoms as those with a date of onset within 1 month from overall COVID-19 onset. In addition, when we considered lung function results as an objective measure of PASC symptoms related to respiratory sequelae, this allowed us to reduce noise from self-reported symptoms. We did not collect pre-COVID serum samples available to distinguish any pre-existing aberrant cytokine levels resulting from underlying comorbidities. Finally, the vast majority of our cohort were infected with the wild-type variant of SARS-CoV-2, limiting the external validity to individuals developing long COVID following infection with currently-circulating sub-variants. However, our findings continue to shed light on the (immuno-) pathogenesis of long COVID which will inform future research among different populations.

## Conclusions

Our study indicates that immune dysregulation is associated with PASC, especially when defining PASC as having impaired pulmonary function. In addition, early raised IL1β levels were strongly predictive of ongoing PASC at 6 months in our analyses. Our findings therefore suggest that immune dysregulation plays an important role in the pathogenesis of ongoing symptoms. An essential further question is if immune dysregulation is causally related to PASC and, if so, what the driving pathological processes may be. Next steps in addressing this question may include focusing on immune dysregulation in individuals with well-defined clusters of PASC symptoms in relation to organ involvement.

## Supporting information

**S1 File.**
(DOCX)

## Acknowledgments

**RECoVERED Study Group**: *From the Public Health Service of Amsterdam*: Ivette Agard, Jane Ayal, Floor Cavdar, Marianne Craanen, Udi Davinovich, Annemarieke Deuring, Annelies van Dijk, Maartje Dijkstra, Ertan Ersan, Laura del Grande, Joost Hartman, Nelleke Koedoot, Romy Lebbink, Tjalling Leenstra, Dominique Loomans, Agata Makowska, Tom du Maine, Ilja de Man, Amy Matser, Lizenka van der Meij, Marleen van Polanen, Maria Oud, Clark Reid, Leeann Storey, Marije de Wit, Marc van Wijk. *From Amsterdam University Medical Centers*: Joyce van Assem, Marijne van Beek, Orlane Figaroa, Leah Frenkel, Agnes Harskamp-Hol-werda, Mette Hazenberg, Soemeja Hidad, Nina de Jong, Hans Knoop, Lara Kuijt, Anja Lok, Eric Moll van Charante, Colin Russell, Annelou van der Veen, Bas Verkaik, Gerben-Rienk Visser.

The authors wish to thank all RECoVERED study participants.

## Author Contributions

**Conceptualization:** Elke Wynberg, Alvin X. Han, Neeltje A. Kootstra, Godelieve J. de Bree.

**Data curation:** Elke Wynberg, Hugo D. G. van Willigen, Anouk Verveen, Neeltje A. Kootstra.

**Formal analysis:** Alvin X. Han, Lisa van Pul, Irma Maurer, Ester M. van Leeuwen.

**Funding acquisition:** Joost G. van den Aardweg, Menno D. de Jong, Pythia Nieuwkerk, Maria Prins, Neeltje A. Kootstra, Godelieve J. de Bree.

**Investigation:** Lisa van Pul, Irma Maurer, Ester M. van Leeuwen, Joost G. van den Aardweg, Neeltje A. Kootstra, Godelieve J. de Bree.

**Methodology:** Elke Wynberg, Alvin X. Han, Lisa van Pul, Irma Maurer, Ester M. van Leeuwen, Joost G. van den Aardweg, Maria Prins, Neeltje A. Kootstra, Godelieve J. de Bree.

**Project administration:** Elke Wynberg, Hugo D. G. van Willigen, Anouk Verveen, Menno D. de Jong, Pythia Nieuwkerk, Maria Prins, Neeltje A. Kootstra, Godelieve J. de Bree.

**Resources:** Menno D. de Jong, Pythia Nieuwkerk, Maria Prins.

**Supervision:** Joost G. van den Aardweg, Menno D. de Jong, Pythia Nieuwkerk, Maria Prins, Neeltje A. Kootstra, Godelieve J. de Bree.

**Validation:** Elke Wynberg, Joost G. van den Aardweg, Neeltje A. Kootstra, Godelieve J. de Bree.

**Visualization:** Alvin X. Han.

**Writing – original draft:** Elke Wynberg, Alvin X. Han, Neeltje A. Kootstra, Godelieve J. de Bree.

**Writing – review & editing:** Elke Wynberg, Alvin X. Han, Hugo D. G. van Willigen, Anouk Verveen, Lisa van Pul, Irma Maurer, Ester M. van Leeuwen, Joost G. van den Aardweg, Menno D. de Jong, Pythia Nieuwkerk, Maria Prins, Neeltje A. Kootstra, Godelieve J. de Bree.

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
