## [Decision Letter · Decision Letter 0]

11 Sep 2023

PONE-D-23-20788Inflammatory profiles are associated with long COVID up to 6 months after illness onset: a prospective cohort study of individuals with mild to critical COVID-19PLOS ONE

Dear Dr. Wynberg,

Thank you for submitting your manuscript to PLOS ONE. After careful consideration, we feel that it has merit but does not fully meet PLOS ONE’s publication criteria as it currently stands. Therefore, we invite you to submit a revised version of the manuscript that addresses the points raised during the review process.

We look forward to receiving your revised manuscript.

Kind regards,

Mickael Essouma, M. D.

Academic Editor

PLOS ONE

Additional Editor Comments:

General comment

You have a good study to provide great insights into the pathogenesis of long COVID as your study is a prospective cohort, which is the best study design in observational epidemiology to inform on disease aetiology. In addition, the assessment of pro-inflammatory cytokines gives you the opportunity to discuss underlying inflammatory mechanisms of long COVID. However, you missed your opportunities with this version of the manuscript. I there make the following specific comments to further help you improve your manuscript:

1-Major comments

The title of the manuscript needs a revision, taking into considerations revisions suggested for the main manuscript. I would say: "Evolution of serum levels of pro-inflammatory cytokines in individuals with long COVID and its predictors: data from a 6-month prospective follow-up of individuals with COVID-19". Same thing for keywords that should be taken from the tile. The abstract will also need revision.

Introduction: Call it Background as recommended in the journal`s policy. I suggest revising it to provide within one page: a definition of long COVID (see https://apps.who.int/iris/bitstream/handle/10665/366126/WHO-2019-nCoV-Post-COVID-19-condition-CA-Clinical-case-definition-2023.1-eng.pdf and https://apps.who.int/iris/bitstream/handle/10665/345824/WHO-2019-nCoV-Post-COVID-19-condition-Clinical-case-definition-2021.1-eng.pdf) as well as its main manifestations (https://doi.org/10.1038/s41598-021-95565-8) in both paediatric and adult populations in the first paragraph, a discussion of the pathophysiology of long COVID highlighting gaps in knowledge notably regarding immune dysregulation: you have these details here https://doi.org/10.1038/s41579-022-00846-2. The third paragraph should then clearly, simply and succinctly state the aim of your study: assess the evolution of serum levels of some pro-inflammatory cytokines in individuals with PASC as from the early period of COVID-19 infection, as well as predictors of PASC, with overarching goal to contribute to the advancement of knowledge on the pathophysiology of PASC.

Methods: please, conform to the STROBE guidelines and state that you conformed to the STROBE guidelines at the beginning of the Methods section. Desired sub-sections of this section include: " Study design, setting and participants (where you describe the RECOVERED study, For the description of the RECOVERED study which is very important, keywords are: when was it launched? It is a cohort study with prospectively collected data, what are inclusion [e.g., age, gender, individuals with mild to critical COVID-19: how do you define mild to critical COVID-19 with references?]and exclusion criteria of the study population? You present data from which period? If the cohort has already been extensively presented elsewhere as it seems, end this brief description with a sentence to orientate the readers towards that/those paper[s]), Data collection (where you describe the data collected [clinical [e.g., BMI] imaging [e.g., HRCT], functional [e.g., DLCO], and cytokines, with emphasis on cytokines: list the cytokines of interest here], how you collected those data with clear specification of time points of data collection, how blood samples for cytokine assessment were transported to the laboratory and conserved before the assessment of cytokines, and how cytokines were assessed in the laboratory: this subsection should be the longest of the methods section), exposition and outcome measures (specify the exposure [COVID], outcome measures [primary: serum levels of pro-inflammatory cytokines; list them at t...], secondary outcomes?), statistical analysis (important elements of this sub-section that should be clearly and succinctly delineated include: where did you store data collected on questionnaires? Did you export them to another software for analysis? Which are the main variables analysed? Which statistical tests did you use to assess the predictors [see https://doi.org/10.1093/ejendo/lvac012 and doi: 10.1093/ndt/gfw459]? Which are the candidate predictors [this is where you need to define the predictors such as high BMI]? For which PASC manifestations do you assess the predictive role of those pro-inflammatory cytokines? How did you define statistical significance of your results?), and ethical issues.

Results and discussion are biased because of methodological biases. Notably, a population without a history COVID-19 is not necessary to assess predictors of PASC as you can read from the articles on predictor assessment provided above. I can understand that you used the blood samples of individuals without a history of COVID-19 as control samples in the laboratory, but this should be clearly

and only explained under the data collection sub-section of the methods section as mentioned above. It should be very clear in the revised methods section as suggested above, that your study population includes only participants from the RECOVERED cohort who were all exposed to COVID-19 infection. Furthermore, you included individuals aged >= 16 years, and disease in adolescents is not exactly the same as in adults (see the literature from the WHO on the definition of post-covid syndrome mentioned above. Some consider paediatric patients up to 21 years. So, it would be interesting to disentangle data from adults to those from adolescents, or in case the sample of adolescents does not allow this because it is too small, acknowledge this issue in the discussion as a limitation. Moreover, it will be interesting to disentangle data from patients who had persistent covid manifestations to those who developed post-covid manifestations after a period free of symptoms of COVID-19. The reviewers have raised a similar comment. The results section would better be split into three sub-sections: general characteristics of the study population, Evolution of serum levels pro-inflammatory cytokines in individuals with PASC [the study population that developed the outcome] compared to the COVID-19 infected population from the RECOVERED that did not develop the PASC outcome throughout the study period (this is where figures on the evolution of pro-inflammatory cytokines need to be linked), and predictors of PASC/PASC manifestations (the manifestations clustered and disentangled).

In the discussion: the first paragraph to summarize succinctly your main findings (regarding the evolution of levels of serum pro-inflammatory cytokines and predictors of PASC/PASC manifestations). Other paragraphs to compare your results with data from the literature, a paragraph to summarize how your study advances knowledges in the pathophysiology of PASC based on inflammatory mechanisms (reflect on potential mechanisms based on data from the literature and your results on pro-inflammatory cytokine assessment: you may add a figure to better highlight the potential mechanisms), and finally a paragraph on strengths and limitations of your study (do not forget to focus on limitations of a prospective cohort study as well as limitations with blood sample collection, conservation and analyses in the laboratory, as these are the limitations that have the potential to affect the interpretation of your study results. Other limitations are not worth mentioning).

The conclusion should be a take-home message with a mention of the perspective of your study for future studies on PASC.

2-Minor comments

Statements (conflicts of interests, authors` contributions, acknowledgments, funding) should come after the conclusion, and before the references section.

Format references according to the journal`s policy. References 1 to 4: I would not mention them as the study is about post-COVID (long COVID) syndrome. Reference 9 needs to be revised or replaced by another well identifiable reference, same thing for all references with print and electronic in brackets. Make sure there is no citation gaming in your manuscript. I will revise it next time.

Revise tables and figures as required

Language editing also needed.

Reviewers' comments:

Reviewer's Responses to Questions

**Comments to the Author**

1. Is the manuscript technically sound, and do the data support the conclusions?

Reviewer #1: Yes

Reviewer #2: Yes

2. Has the statistical analysis been performed appropriately and rigorously? 

Reviewer #1: Yes

Reviewer #2: Yes

3. Have the authors made all data underlying the findings in their manuscript fully available?

Reviewer #1: Yes

Reviewer #2: Yes

4. Is the manuscript presented in an intelligible fashion and written in standard English?

Reviewer #1: Yes

Reviewer #2: Yes

5. Review Comments to the Author

Reviewer #1: The manuscript analyze the relationship between cytokine profiles and persistent COVID-19. The authors compares cytokine levels in a follow-up cohort of COVID-19 patients recovered from acute infection. They explore if there is a relation between cytokine profile and the presence of persistent COVID-19 assessed with clinical questionnaires. They use healthy non-infected individuals as a controls for cytokine profiles.

The paper includes a limited number of patients (n=186), with a total of 101 patients reporting persistent COVID-19.

Interesting results may be outlined, as the relation between persistent COVID-19 and CRP levels and in the follow-up period and the early IL-1b levels as a predictor of persistent COVID-19.

Multicentric prospective study with a systematic work and a nice and detailed presentation of the results.

Reviewer #2: In this manuscript, Wynberg et al. investigated the role of immune dysregulation in the development of Post-acute sequelae of SARS-CoV-2. They analyzed pro and anti-inflammatory cytokine levels in serum longitudinally for 6 months to establish the correlation with PASC. The work is well done, but there are a few comments that should be addressed:

Did the authors investigate viral genes or infectious virus? It has been reported that persistent viral RNA could prolong the inflammatory response leading to long COVID.

Previous articles (PMID: 35846757, PMID: 35732153) show elevated TNF-a in long COVID patients. The disparity in data should be discussed.

Did the authors compare in the cytokines level in presence and absence of the vaccine? Did the authors analyze data in different strains of SARS-CoV-2?

6. PLOS authors have the option to publish the peer review history of their article (what does this mean?). If published, this will include your full peer review and any attached files.

Reviewer #1: No

Reviewer #2: No

---

## [Decision Letter · Decision Letter 1]

27 Nov 2023

PONE-D-23-20788R1Inflammatory profiles are associated with long COVID up to 6 months after illness onset: a prospective cohort study of individuals with mild to critical COVID-19PLOS ONE

Dear Dr. Wynberg,

Thank you for submitting your manuscript to PLOS ONE. After careful consideration, we feel that it has merit but does not fully meet PLOS ONE’s publication criteria as it currently stands. Therefore, we invite you to submit a revised version of the manuscript that addresses the points raised during the review process.

We look forward to receiving your revised manuscript.

Kind regards,

Mickael Essouma, M. D.

Academic Editor

PLOS ONE

Journal Requirements:

Additional Editor Comments:

The manuscript has been improved. There are still important rooms for improvement.

The title needs to be free of ambiguity: "Inflammatory profiles are associated with long COVID up to 6 months after mild to critical COVID-19: data from the RECOVERED cohort"

Introduction: the second paragraph needs to be reduced. I would mention the current pathophysiologic hypotheses for long COVID (viral persistence, chronic inflammation, hypercoagulability and autonomic dysfunction: see https://doi.org/10.1038/s41577-023-00966-7) and then briefly describe what is known and unknown about the chronic inflammation theory (because your study dealt with the chronic inflammation theory). The third paragraph also needs to be reduced. You could say something like: "The aim of our study was to further explore the chronic inflammation theory about the pathophysiology of long COVID through a longitudinal assessment of inflammatory cytokines present in individuals with COVID-19 at disease onset."

Methods

Page 7 line 100: it will be helpful for the readers to see the reference for STROBE guidelines.

Line 103: before describing RECOVERED, you would like to state something like: "This is a subset of the RECOVERED study."

"We understand that a COVID-negative control group is not necessary to assess the impact of PASC on cytokine concentrations. However, for clinical relevance it is not only useful to know the relative

effect of PASC on cytokine concentrations but also whether the levels reported are of clinical significance. This can only be determined by observing cytokine levels in an uninfected, healthy reference population." I still do not see the importance of that group. You start the study already knowing the physiologic and abnormal levels of the serum cytokines you assessed in the study. You only need to mention the healthy individuals as laboratory controls when reporting about laboratory assay of serum cytokines. THERE IS NO CONTROL GROUP IN A COHORT STUDY. THE COHORT STUDY HAS TWO STUDY POPULATIONS: THOSE WHO GO ON TO DEVELOP THE DISEASE OF INTEREST (LONG COVID FOR THIS STUDY) AFTER CONTACT WITH THE EXPOSURE (COVID-19 FOR THIS STUDY), AND THOSE WHO DO NOT DEVELOP THE DISEASE (FOR THIS STUDY: THOSE WHO HAD covid-19 AND DID NOT DEVELOP LONG COVID).

After the sub-section "Study design and participant enrolment", please mention a section " Data collection " under which you detail the clinical and laboratory (do not forget to specify the cytokines assessed [ideally classified by families such as IFN-related cytokines, chemokines, TNF cytokines, interleukins: I am surprised I do not see IL-6 a very important pro-inflammatory cytokine on line 140] and say why you assessed specifically those cytokines since there are many pro-inflammatory cytokines; CD14 and CD163 are monocyte/macrophage surface markers, not cytokines I therefore do not understand why they are reported and CRP on line 138) data collected, the methods of data collection used, and the timings of data collection (to, at 3 months, at 6 months). So, this sub-section will contain information from the line 119 to the line 145. The title states that the total duration of the follow up for each participant was six months, so I do not understand why on line 131 you mention that you collected blood samples at 12, 18 , 24 months. After the data collection sub-section, add a sub-section termed "Outcome" which is very important for a cohort. The main outcome here being the serum levels of pro-inflammatory cytokines at 6 months post-COVID infection. After the outcome sub-section, then follow the definitions and statistical analysis sub-sections as you did.

Line 153 "COVID-19 clinical severity was categorised according to WHO criteria[20]". Please, describe those grades of severity.

Lines 154-156: "Diffusion capacity (DLCO) at 6 months after illness onset was defined as impaired according to American Thoracic Society (ATS) European Respiratory Society guidelines[21], as described previously described[19]". Is this not a secondary outcome of the study? If it is the case, then do not forget to mention that in the outcome sub-section.

Lines 159-161: "Socio-demographic, clinical and study characteristics were compared between included participants with and without PASC at 12 weeks after illness onset. To assess selection bias, we compared features of included and excluded participants." Please, delete.

Generally speaking, your statistical analysis sub-section is long, difficult to read and confusing. Please, state the software where you stored the data collected, whether you needed to export data to another software for statistical analysis, the statistical tests used for the different study outcomes, the statistical tests used to assess the factors associated with the study outcomes, the candidate factors

associated with the different outcomes as assessed in the statistical models, and the definition of statistical significance you used.

A cohort based on STROBE should start with the sub-section " Baseline characteristics of the study population", Then follow the sub-sections "primary outcome(s)" and "secondary outcomes". Then you can end the results section with the sub-section " factors associated with persistence of a pro-inflammatory profile in individuals with COVID-19".

Discussion: last time, I proposed a discussion plan. Please, stick to it. The paragraph about study's strengths and limitations should be shortened: regarding limitations, please stick to comments about the shortcomings of a 6-month prospective cohort study.

Conclusion

Line 380 "In summary" needs to be deleted. The conclusion should match with the main objective stated in the introduction, and call for further studies on the topic of your study.

Ultimately, the abstract will also need a revision, and should not exceed 300 words. The keywords should contain terms from the title and eventually the abstract.

Reviewers' comments:

Reviewer's Responses to Questions

**Comments to the Author**

1. If the authors have adequately addressed your comments raised in a previous round of review and you feel that this manuscript is now acceptable for publication, you may indicate that here to bypass the “Comments to the Author” section, enter your conflict of interest statement in the “Confidential to Editor” section, and submit your "Accept" recommendation.

Reviewer #2: All comments have been addressed

2. Is the manuscript technically sound, and do the data support the conclusions?

Reviewer #2: Yes

3. Has the statistical analysis been performed appropriately and rigorously? 

Reviewer #2: Yes

4. Have the authors made all data underlying the findings in their manuscript fully available?

Reviewer #2: Yes

5. Is the manuscript presented in an intelligible fashion and written in standard English?

Reviewer #2: Yes

6. Review Comments to the Author

Reviewer #2: (No Response)

7. PLOS authors have the option to publish the peer review history of their article (what does this mean?). If published, this will include your full peer review and any attached files.

Reviewer #2: No

---

## [Author Response · Author response to Decision Letter 1]

2 Feb 2024

Dear Dr. Mickael Essouma,

Thank you for taking the time to review our article.

Please see our response to each specific comment in the attached file.

---

## [Editor Report · Decision Letter 2]

12 Feb 2024

PONE-D-23-20788R2Inflammatory profiles are associated with long COVID up to 6 months after illness onset: a prospective cohort study of individuals with mild to critical COVID-19PLOS ONE

Dear Dr. Wynberg,

Thank you for submitting your manuscript to PLOS ONE. After careful consideration, we feel that it has merit but does not fully meet PLOS ONE’s publication criteria as it currently stands. Therefore, we invite you to submit a revised version of the manuscript that addresses the points raised during the review process.

We look forward to receiving your revised manuscript.

Kind regards,

Mickael Essouma, M. D.

Academic Editor

PLOS ONE

Journal Requirements:

**Additional Editor Comments:**

The manuscript has been improved, but there are some revisions that need to be made so that the published report matches the level of field work made.

Ref 16 should be revised.

Tables should come at the end of the full-text manuscript, to facilitate the assessment of the manuscript.

Lines 100-102: the citation of the WHO is needed.

Line 136: the reference provided is not right. Is there a link with reference 18 cited in line 160?

Lines 153 and 158: 21 or 20?

I advise against citing preprints. If you have to cite a preprint, specify that the reference is a preprint.

Line 175: consider writing "non-specific antinuclear antibodies" instead of "total antinuclear antibodies". Line 176: please rephrase "as indicators of possible auto-immunity" as "as markers of autoimmunity". Indeed, ANA are the best markers of autoimmunity.

Lines 182-183. I suggesting deleting "when assessing possible early biomarkers for later PASC" as it brings confusion to the message you want to give.

Line 191: ref [14] is not the WHO reference.

Line 198 American Thoracic Society European Respiratory Society should read American Thoracic Society/European Respiratory Society.

Line 199: "described" should be deleted.

Lines 204 and 205: "To assess selection bias, we compared features of included and excluded participants." This sentence is not right and should be deleted. In fact, all the paragraph in lines 203-207 is not necessary. The paragraph in lines 233-240 is difficult to understand. Are the early predictors of PASC you are talking about different from the pre-COVID and COVID-related factors you are talking about in the paragraph in lines 219-231? You say that you have measured the levels of non-specific ANA within 4 weeks of COVID onset. However, I do not see how you used that data in the predictor analysis. When describing the statistical analysis, focus on the description of: 1) The statistical tests used and specify the variables that were analysed using those statistical tests. Here again, focus on the study outcomes so that readers will be able to have similar results using those statistical tests. This is why I said that the paragraph in lines 203-207 is not necessary. 2) The definition of statistical significance for the outcomes of interest, 3) then how you reported the results. 4) finally, the software where you stored the data before the analysis and how you exported the data from that software to the statistical analysis software. It reads something like this:

"We determined box and whisker plots of serum concentrations of cytokines over time. During this analysis, data of study participants with and without PASC at 12 and 24 weeks after COVID-onset were compared with data from the uninfected healthy control population using the Mann-Whitney and Bonferroni correction tests. We segregated serum cytokine levels at 21-24 weeks post-COVID by DLCO status (normal vs abnormal). We subsequently used the linear mixed-effects test to assess the effect of pre-COVID and COVID-related factors on serum concentrations of cytokines at 12 and 24 weeks post-COVID. The pre-COVID factors assessed included.... [reference important to show that you chose them based on the literature]. The COVID-related factors assessed included ....[reference again]. We used a correlation matrix to help assess the reciprocal associations between cytokines at 9-12 and 21-24 weeks post-COVID. Add a comprehensive synthesis of the paragraph in lines 233-240 here if this paragraph is still necessary.

All data were collected and stored in a secure database and exported for statistical analyses to Python. Analyses were performed using the statsmodels package (v. 0.13.2)[21], whilst the random forest regression analyses were performed in Python using the scikit-learn package (v.2441.1.3)."

Paragraph in lines 248-255: state the specific numbers, avoid the adverbs.

Table 1 is confusing and needs to be revised because it is supposed to display the baseline characteristics, but I see the 3-month data in that table. Follow-up data should not be in that table. I also do not understand why there is a word recovered in that table whereas study participants belong to the RECOVERED cohort. This may be confusing for readers. The table is unacceptably long. A shorter title such as "Baseline characteristics of study participants with and without PASC after COVID-19 infection" would be better. Then, you see that the relevant comparisons is that between the two study populations reported in the table's title. If you want to show the baseline characteristics of the other study population, do this in tables that you send to the supplementary material. Variables in table 1 should be those that you found relevant predictors of PASC, along with demographic characteristics. I would organize its arrows like this: demographic (age, sex, level of education) and epidemiological (e.g., COVID-19 vaccination status, comorbidities) data, followed by clinical data (symptomatic/asymptomatic at diagnosis, cardiorespiratory features, disease severity: mild, moderate, critical), and then management data (ambulatory/hospital/ICU care, drugs used: steroids, other immunomodulatory and immunosuppressive drugs). I would not mention the migration status. You can also merge all the comorbidities in a unique variable "comorbidities". I am happy to see the vaccination status in table 1, but I do not see it among the candidate predictors of PASC in the methods section. Please, make a comment about this in the statistical analysis sub-section of the Methods section.

I suggest deleting the text in lines 274-276 which may be confusing. It is already clear in the methods that you used reference samples in the laboratory when assessing serum levels of cytokines.

Lines 277-363: there should be consistency between the way you report these results and the report on study methods. For example, I see "in univariate analysis", "in multivariate analysis", but I have not seen that in the statistical analysis sub-section. You report that determinants of serum cytokine levels are part of the primary outcome, but this is not what I have read in the methods section. There is this problem of whether you make a difference between early predictors of PASC and pre-COVID and COVID-related factors highlighted above that is also apparent here. Instead of saying after "illness onset", state "after COVID-19 onset" throughout the text. Focus this report of results on data from participants with and without PASC after COVID-19 please. No need to emphasize on the population from which you collected reference samples. When you state in the text that this result is avilable in supplementary tables...Readers will themselves go and find in those tables results of the population from which you collected reference samples. This is the most crucial part of the result section. Write it in the simplest and most intelligible way referring readers to the figures that are very beautiful Draw our attention to the figures as much as you can.

A more exact name of Figure 1 is "Box and Whisker plots of serum cytokine levels in the study population at 9-12 and 21-24 weeks post-COVID". I can not assess the other figures until you resolve the inconsistency across the methods section and lines 277-363 of the results section.

Lines 366-367: do you want to say that you assessed the evolution of cytokines in individuals with PASC up to 6 months (24 weeks) post-COVID? The study objective, methods, results and discussion should be consistent. Please, take your time to fix these issues before submitting the revised version of the manuscript. You have strong results. Strengthen your discussion. The first paragraph of the discussion should summarize the study results in a way that is consistent with the objective stated in the introduction, as well as methods and results sections. The second paragraph should explain the results of Figure 1 taking into consideration data from the literature, and you know that this is an ever-evolving field. So, try as much as possible to have the latest evidence when making this discussion. In the subsequent paragraph, explain your results on predictors of high levels of pro-inflammatory serum cytokines and how those predictors and the pro-inflammatory cytokines could influence the pathogenesis of PASC.

Then discuss the implications of your research on this currently very famous research area. How do you advance our knowledge? What remains to be done to further elucidate the hypotheses you elaborated while interpreting your results. Finally, state the limitations and strengths of your study.

I am surprised that you measure non-specific ANA, and you did not report fhe titers of ANA in your study population (I would put them among clinical data in table 1. COVID-19 is a confirmed risk factor of autoimmune diseases. It is not clear whether autoimmunity is involved in the pathogenesis of PASC. I thought that may be you will provide an insight into the involvement of autoimmunity in the pathogenesis of PASc, especially since you know that interferons (e.g. IP-10) are involved in the pathogenesis of autoimmune diseases and autoantibody production s a downstream reaction of interferon signaling.

It is in the conclusion that it becomes very clear that you defined PASC as impaired DLCo in some circumstances. This is not so clear in the methods (lines 186-199) and results section, unfortunately. Clearly state in the methods section. The conclusion should be strong. Waht message would you want to remember if you forgot everything about your manuscript? Put that message in the conclusion. I guess it will fit with the study's main objective.

Format references according to the PLOS ONE policy.

---

## [Author Response · Author response to Decision Letter 2]

16 May 2024

Reviewer’s / editorial comments:

The manuscript has been improved, but there are some revisions that need to be made so that the published report matches the level of field work made.

Ref 16 should be revised: it is unclear how this reference should be revised, this reference refers to the STROBE guideline and checklist that is available on the mentioned website.

Tables should come at the end of the full-text manuscript, to facilitate the assessment of the manuscript: following this comment, table 1 is moved to the end of the manuscript, see p23

Lines 100-102: the citation of the WHO is needed: this reference was added

Line 136: the reference provided is not right. Is there a link with reference 18 cited in line 160? indeed reference 1 is incorrect: the right reference is indeed no 18: this is changed accordingly

Lines 153 and 158: 21 or 20? We are not sure where these numbers refer to, I cannot find them in the mentioned lines.

I advise against citing preprints. If you have to cite a preprint, specify that the reference is a preprint: we checked the manuscript for references to pre-prints and updated this (ref 15) to the publication 

Line 175: consider writing "non-specific antinuclear antibodies" instead of "total antinuclear antibodies": at this point we disagree with the reviewer: antinuclear antibodies encompass specific antibodies (for instance adsDNA) and therefore cannot be named “non-specific”. 

Line 176: please rephrase "as indicators of possible auto-immunity" as "as markers of autoimmunity". Indeed, ANA are the best markers of autoimmunity: this was changed accordingly.

Lines 182-183. I suggesting deleting "when assessing possible early biomarkers for later PASC" as it brings confusion to the message you want to give: this was changed accordingly.

Line 191: ref [14] is not the WHO reference. this was changed accordingly.

Line 198 American Thoracic Society European Respiratory Society should read American Thoracic Society/European Respiratory Society: this was changed accordingly.

Line 199: "described" should be deleted: this was changed accordingly.

Lines 204 and 205: "To assess selection bias, we compared features of included and excluded participants." This sentence is not right and should be deleted. In fact, all the paragraph in lines 203-207 is not necessary: in our opinion ot is relevant to assess selection bias therefore we decided to leave this section in (of note selection bias data are presented in table S1 and in the first section of the results paragraph).

The paragraph in lines 233-240 is difficult to understand. Are the early predictors of PASC you are talking about different from the pre-COVID and COVID-related factors you are talking about in the paragraph in lines 219-231? With earlier predictors of PASC we mean predicting factors at illness onset. This was clarified in the text (see l239).

You say that you have measured the levels of non-specific ANA within 4 weeks of COVID onset. However, I do not see how you used that data in the predictor analysis: please refer to results section lines 344- 346 and figure 4A: here we list the data on ANA as predictive factor for PASC. 

When describing the statistical analysis, focus on the description of: 1) The statistical tests used and specify the variables that were analysed using those statistical tests. Here again, focus on the study outcomes so that readers will be able to have similar results using those statistical tests. This is why I said that the paragraph in lines 203-207 is not necessary. 2) The definition of statistical significance for the outcomes of interest, 3) then how you reported the results. 4) finally, the software where you stored the data before the analysis and how you exported the data from that software to the statistical analysis software. It reads something like this: 

"We determined box and whisker plots of serum concentrations of cytokines over time. During this analysis, data of study participants with and without PASC at 12 and 24 weeks after COVID-onset were compared with data from the uninfected healthy control population using the Mann-Whitney and Bonferroni correction tests. We segregated serum cytokine levels at 21-24 weeks post-COVID by DLCO status (normal vs abnormal). We subsequently used the linear mixed-effects test to assess the effect of pre-COVID and COVID-related factors on serum concentrations of cytokines at 12 and 24 weeks post-COVID. The pre-COVID factors assessed included.... [reference important to show that you chose them based on the literature]. The COVID-related factors assessed included ....[reference again]. We used a correlation matrix to help assess the reciprocal associations between cytokines at 9-12 and 21-24 weeks post-COVID. Add a comprehensive synthesis of the paragraph in lines 233-240 here if this paragraph is still necessary. All data were collected and stored in a secure database and exported for statistical analyses to Python. Analyses were performed using the statsmodels package (v. 0.13.2)[21], whilst the random forest regression analyses were performed in Python using the scikit-learn package (v.2441.1.3).": in our opinion the statics paragraph does explain in a logical order the test used and is displayed in analogy to the results section. However, we added the suggestions of the reviewer and also more clearly highlighted which sections belong to the different study outcomes. 

Paragraph in lines 248-255: state the specific numbers, avoid the adverbs: I am unsure what the reviewer means, in this section we do list the numbers and I do not see irrelevant adverbs. 

Table 1 is confusing and needs to be revised because it is supposed to display the baseline characteristics, but I see the 3-month data in that table: this may be a misunderstanding: table 1 does show the baseline characteristics (second column) and the baseline characteristics of groups stratified for participants that later in the study develop PASC (third and fourth column). To clarify this information was added to the legend of the table. 

Follow-up data should not be in that table: please refer to our previous answer

I also do not understand why there is a word recovered in that table whereas study participants belong to the RECOVERED cohort. This may be confusing for readers: it is difficult to find a proper synonym for recovered (from illness symptoms). We changed it for now in restored symptoms and ongoing symptoms.

The table is unacceptably long. A shorter title such as "Baseline characteristics of study participants with and without PASC after COVID-19 infection" would be better: this was changed accordingly

Then, you see that the relevant comparisons is that between the two study populations reported in the table's title. If you want to show the baseline characteristics of the other study population, do this in tables that you send to the supplementary material. Variables in table 1 should be those that you found relevant predictors of PASC, along with demographic characteristics: according to the comment the table 1 was shortened and the following data: sex, BMI, initial disease severity, vaccination and demographic predictors were included. All other baseline data that we consider relevant in order to have insight in the study population characteristics were moved to a new supplementary table (s2).

I would organize its arrows like this: demographic (age, sex, level of education) and epidemiological (e.g., COVID-19 vaccination status, comorbidities) data, followed by clinical data (symptomatic/asymptomatic at diagnosis, cardiorespiratory features, disease severity: mild, moderate, critical), and then management data (ambulatory/hospital/ICU care, drugs used: steroids, other immunomodulatory and immunosuppressive drugs). I would not mention the migration status. You can also merge all the comorbidities in a unique variable "comorbidities": since the data displayed in the table are no re-arranged (see previous answer) some of the here mentioned factors are now in suppl table 2.

 I am happy to see the vaccination status in table 1, but I do not see it among the candidate predictors of PASC in the methods section. Please, make a comment about this in the statistical analysis sub-section of the Methods section: vaccination is already mentioned in the statistics section: see lines 230 and 231.

I suggest deleting the text in lines 274-276 which may be confusing. It is already clear in the methods that you used reference samples in the laboratory when assessing serum levels of cytokines: we suggest to leave this in since the text here (in the current revised ms (v3) l261-263) lists the characteristics of the reference samples which belong in the results section.

Lines 277-363: there should be consistency between the way you report these results and the report on study methods. For example, I see "in univariate analysis", "in multivariate analysis", but I have not seen that in the statistical analysis sub-section: the method used for the multivariate analysis in the results section is the linear mixed effect model as described in the methods / statistics section. We clarified this in the revised version of the statistics section.

You report that determinants of serum cytokine levels are part of the primary outcome, but this is not what I have read in the methods section: this was added to the methods section (l212).

There is this problem of whether you make a difference between early predictors of PASC and pre-COVID and COVID-related factors highlighted above that is also apparent here. Instead of saying after "illness onset", state "after COVID-19 onset" throughout the text: accordingly, illness onset is replaced by COVID-19 onset throughout the text of the manuscript.

Focus this report of results on data from participants with and without PASC after COVID-19 please. No need to emphasize on the population from which you collected reference samples. When you state in the text that this result is available in supplementary tables...Readers will themselves go and find in those tables results of the population from which you collected reference samples. This is the most crucial part of the result section. Write it in the simplest and most intelligible way referring readers to the figures that are very beautiful Draw our attention to the figures as much as you can: we thank the reviewer for this compliment

A more exact name of Figure 1 is "Box and Whisker plots of serum cytokine levels in the study population at 9-12 and 21-24 weeks post-COVID": the title was changed accordingly 

I can not assess the other figures until you resolve the inconsistency across the methods section and lines 277-363 of the results section: please refer to our answer at the top of page 2.

Lines 366-367: do you want to say that you assessed the evolution of cytokines in individuals with PASC up to 6 months (24 weeks) post-COVID?: indeed, that was also stated as such in the abstract: “We described longitudinal trajectories of cytokines in adults up to 6 months following SARS-CoV-2 infection and explored early predictors of PASC”. 

The study objective, methods, results and discussion should be consistent. Please, take your time to fix these issues before submitting the revised version of the manuscript. You have strong results. Strengthen your discussion. The first paragraph of the discussion should summarize the study results in a way that is consistent with the objective stated in the introduction, as well as methods and results sections: we re-read this first paragraph and checked the consistence with the abstract as well as our primary and secondary aims and feel that as the text is now there is consistence throughout the manuscript. The reviewer suggests to “strengthen the discussion”, if that is indeed needed then specific suggestions would be welcome.

The second paragraph should explain the results of Figure 1 taking into consideration data from the literature, and you know that this is an ever-evolving field. So, try as much as possible to have the latest evidence when making this discussion: following this comment a few recent landmark papers were added as reference (Peluso Semin Imm 2024; Yin Nat Imm 2023; Thaweethai JAMA 2023).

In the subsequent paragraph, explain your results on predictors of high levels of pro-inflammatory serum cytokines and how those predictors and the pro-inflammatory cytokines could influence the pathogenesis of PASC: we added this to the discussion (lines 492-498).

Then discuss the implications of your research on this currently very famous research area. How do you advance our knowledge? What remains to be done to further elucidate the hypotheses you elaborated while interpreting your results: this was added to the final part of the conclusion (lines 528-532)

Finally, state the limitations and strengths of your study: We think the reviewer has overlooked the limitation as listed in the discussion see lines 508-520

I am surprised that you measure non-specific ANA, and you did not report the titers of ANA in your study population (I would put them among clinical data in table 1. COVID-19 is a confirmed risk factor of autoimmune diseases. It is not clear whether autoimmunity is involved in the pathogenesis of PASC. I thought that may be you will provide an insight into the involvement of autoimmunity in the pathogenesis of PASc, especially since you know that interferons (e.g. IP- 10) are involved in the pathogenesis of autoimmune diseases and autoantibody production s a downstream reaction of interferon signaling: we agree with the reviewer that there are several studies that show a potential role for auto-immunity in development of PASC. There are however different categories of autoimmunity that each have a different pathophysiological background (Bodansky JCI Insight 2023, Peluso CID 2022, ). ANA abs for instance are a different category than antibodies against type 1 IFNs or cytokines. It was beyond the scope of our study to generate a comprehensive analysis of all types of auto-antibodies and we only included ANA abs. Following the suggestion, we added the titers of ANA to table 1. 

It is in the conclusion that it becomes very clear that you defined PASC as impaired DLCo in some circumstances. This is not so clear in the methods (lines 186-199) and results section, unfortunately. Clearly state in the methods section: the suggestion that we define PASC by impaired DLCo seems a misinterpretation of the reviewer. We define PASC based on self-reported symptoms (l 159 ev “PASC was defined as reporting at least one COVID-19 symptom that, from COVID-19 onset, occurred within one month and continued beyond 12 weeks”). The section lines 383-385 in the results section may have been confusing: “when substituting PASC status for impaired diffusion capacity, we found that impaired diffusion capacity was associated with higher levels of CRP, IL6, TNFα, IP10, IL10 and IL17 in multivariable analyses.”. For clarification we reformulated this sentence and omitted PASC status (see adjustment in results l383). Finally, we reformulated lines 514-516 in the discussion to better align with our previous explanation of the approach taken.

The conclusion should be strong. What message would you want to remember if you forgot everything about your manuscript? Put that message in the conclusion. I guess it will fit with the study's main objective: see adjustment in the conclusion (l534-538).

Format references according to the PLOS ONE policy: the references in text and reference list were adjusted.

---

## [Editor Report · Decision Letter 3]

22 May 2024

Inflammatory profiles are associated with long COVID up to 6 months after COVID-19 onset: a prospective cohort study of individuals with mild to critical COVID-19

PONE-D-23-20788R3

Dear Dr. Wynberg,

We’re pleased to inform you that your manuscript has been judged scientifically suitable for publication and will be formally accepted for publication once it meets all outstanding technical requirements.

Kind regards,

Mickael Essouma, M. D.

Academic Editor

PLOS ONE

Additional Editor Comments (optional):

I am unable to open the website (https://www.strobe478statement.org/checklists/) provided with reference 17.

Line 324: consider replacing ref 20 with ref 24 and removing the appended comment in the red box.

Consider removing the comment appended to line 400.
---

## [Editor Report · Acceptance letter]

27 May 2024

PONE-D-23-20788R3 

PLOS ONE

Dear Dr. Wynberg, 

I'm pleased to inform you that your manuscript has been deemed suitable for publication in PLOS ONE. Congratulations! Your manuscript is now being handed over to our production team.

Kind regards, 

on behalf of

Dr. Mickael Essouma 

Academic Editor

PLOS ONE